# Mitochondrial Mechanisms of Neuromuscular Junction Degeneration with Aging

**DOI:** 10.3390/cells9010197

**Published:** 2020-01-13

**Authors:** Maria-Eleni Anagnostou, Russell T. Hepple

**Affiliations:** Department of Physical Therapy and Department of Physiology & Functional Genomics, University of Florida, Gainesville, FL 32608, USA; managnos@usc.edu

**Keywords:** neuromuscular junction, aging, muscle atrophy, mitochondria, motoneuron, skeletal muscle

## Abstract

Skeletal muscle deteriorates with aging, contributing to physical frailty, poor health outcomes, and increased risk of mortality. Denervation is a major driver of changes in aging muscle. This occurs through transient denervation-reinnervation events throughout the aging process that remodel the spatial domain of motor units and alter fiber type. In advanced age, reinnervation wanes, leading to persistent denervation that accelerates muscle atrophy and impaired muscle contractility. Alterations in the muscle fibers and motoneurons are both likely involved in driving denervation through destabilization of the neuromuscular junction. In this respect, mitochondria are implicated in aging and age-related neurodegenerative disorders, and are also likely key to aging muscle changes through their direct effects in muscle fibers and through secondary effects mediated by mitochondrial impairments in motoneurons. Indeed, the large abundance of mitochondria in muscle fibers and motoneurons, that are further concentrated on both sides of the neuromuscular junction, likely renders the neuromuscular junction especially vulnerable to age-related mitochondrial dysfunction. Manifestations of mitochondrial dysfunction with aging include impaired respiratory function, elevated reactive oxygen species production, and increased susceptibility to permeability transition, contributing to reduced ATP generating capacity, oxidative damage, and apoptotic signaling, respectively. Using this framework, in this review we summarize our current knowledge, and relevant gaps, concerning the potential impact of mitochondrial impairment on the aging neuromuscular junction, and the mechanisms involved.

## 1. Introduction

Aging skeletal muscle is characterized by progressive muscle atrophy and a decline in contractile function, which negatively influences quality of life in elderly individuals and predisposes them to increased risk of falls, morbidity and mortality [1,2]. Muscle fiber denervation is among the most influential events affecting aging muscle. This occurs first through transient denervation-reinnervation events that occur throughout adult life and drive changes such as a fiber type shift and type-grouping, and result in an involution of the spatial distribution of muscle fibers within individual motor units (motor unit = a motoneuron and the muscle fibers it innervates) [3]. These changes contribute to impair motor control [4] and reduce contraction-induced increases in muscle blood flow [5] with aging. Denervation has additional adverse impact in advanced age (i.e., ≥75 y in humans), where it contributes to mobility impairment [6] as denervation becomes persistent [7] and causes an acceleration of muscle atrophy [8] and muscle contractile dysfunction [9]. As such, a key to devising strategies to address aging muscle impairment is establishing the mechanisms that lead to denervation of muscle fibers with aging. Alterations in the muscle fibers, the innervating motoneuron, and the specialized synapse between these structures known as the neuromuscular junction, may each play a role in the denervation of skeletal muscle with aging. In considering common mechanisms that could affect these components with aging, mitochondria are frequently implicated as playing a central role in aging. Notably, mitochondria are plentiful in neurons and muscle fibers, and particularly enriched on both sides of the neuromuscular junction [10]. Thus, dysfunction of mitochondria with aging could provide a key explanation for neuromuscular junction instability leading to denervation and resulting adverse impact on aging skeletal muscle. On this basis, we will review the changes in mitochondrial function with aging in motoneurons, the neuromuscular junction and muscle fibers, and potential mechanisms responsible. We will also highlight gaps in knowledge requiring further interrogation.

## 2. Neuromuscular Junction, Denervation-Reinnervation and Atrophy of Aging Muscle

Conditions in which the neuromuscular junction undergoes degeneration precipitate severe muscle atrophy. In these conditions, retraction of motoneuron terminals with or without death of the entire motoneuron causes denervation of individual neuromuscular junctions. Denervation occurs cyclically, where a given denervation event is usually quickly followed by reinnervation by the original motoneuron (if it remains viable) or through collateral sprouting of an adjacent motoneuron axon (if original motoneuron is lost). Importantly, severe muscle atrophy is temporally delayed from the initiation of the cycles of neuromuscular junction degeneration and is consequent to failure of reinnervation. Examples of diseases affecting the neuromuscular junction and which have rapidly evolving atrophy include amyotrophic lateral sclerosis (ALS) and the spinal muscular atrophies (SMA). Mouse models of ALS and SMA both show that neuromuscular junction alterations occur several weeks before onset of clinical symptoms [11,12]. At the other extreme, normal aging involves a slowly evolving muscle atrophy. Similar to ALS and SMA, neuromuscular junction alterations with aging precede muscle atrophy [13], and severe muscle impact is a delayed outcome [9,14]. In this paradigm, the timing of the failure of reinnervation, and thus onset of severe muscle atrophy, depends upon the frequency of neuromuscular junction denervation (higher rate of denervation = faster depletion of reinnervation).

Among the earliest changes to be seen in aging muscle are alterations in neuromuscular junction morphology. Alteration of neuromuscular junction morphology was first reported in aged rodents in 1966 [15] and was subsequently confirmed in elderly humans in multiple studies [16,17,18]. Among the most frequently observed changes in neuromuscular junction morphology, studies have shown changes in the complexity and morphology of the pre-and post-synaptic regions, with an increase in cluster number and reduced cluster size of acetylcholine receptors (AChRs) per endplate (known as AChR cluster fragmentation; Figure 1B), and degeneration of the motoneuron terminals that in the extreme causes denervation of the endplate [19]. In the absence of complete denervation, many of the changes in neuromuscular junction morphology appear to represent compensatory plasticity rather than a functionally inferior structure. For example, in examining the neuromuscular junctions of aged mice, no difference in compound muscle action potentials was observed between endplates with high versus low AChR fragmentation [20]. Regardless of this point, endplates that lack an innervating motoneuron (Figure 1C) are not functional, and with aging there is a gradual accumulation of persistently denervated muscle fibers [21,22] that becomes severe in advanced age in both rodents [7] and humans [23] (Figure 1D).

To provide an example of the evidence for denervation in aging muscle, we have previously noted that very old muscle is characterized by a large abundance of muscle fibers positive for the denervation-responsive sodium channel, Nav_1.5_ (Figure 2A). Remarkably, Nav_1.5_ positive fibers (= denervated) are significantly atrophied versus young adult muscle fibers where no Nav_1.5_ positive fibers are seen. In contrast, Nav_1.5_ negative fibers in very old muscle are not significantly different in size from young adult (Figure 2B), indicating that denervation is the primary cause of atrophy in advanced age. This accumulation of denervated muscle fibers in advanced age is coincident with a dramatic elevation of denervation-responsive transcripts that suggests reinnervation is failing [24] (Figure 2C). Interestingly, in populations that diverge in severity of physical decline in advanced age, a key determining factor appears to be the capacity for reinnervation. Specifically, it has been shown previously that high functioning elderly have more muscle fibers per motor unit [25] and a greater extent of fiber type grouping [23,26], both of which suggest a greater fidelity of reinnervation following denervation in high functioning individuals. Consistent with this idea, we recently found greater fiber type grouping in high functioning elderly women was associated with elevated expression of the reinnervation-promoting cytokine, *fibroblast growth factor binding protein-1* [23]. Collectively, this evidence suggests that understanding the mechanisms causing muscle fiber denervation events throughout adulthood, and the failure of reinnervation in advanced age, is key to developing strategies for delaying the acceleration of muscle atrophy that precipitates the most severe clinical impacts in advanced age.

## 3. Mitochondrial Alterations with Aging

Mitochondria are not only the major energy powerhouse for the cell, but also key regulators of reactive oxygen species (ROS) signaling, Ca^2+^ homeostasis, and the intrinsic pathway of apoptosis. For this reason, mitochondrial dysfunction is considered a central mechanism contributing to atrophy and dysfunction of skeletal muscle with aging [27]. Although impaired mitochondrial function is implicated in degeneration of aging muscle, the majority of research to date has focused upon changes within the muscle fibers, and generally, our thinking about how changes in mitochondria within the motoneurons impact aging muscle is less evolved. Indeed, most of our understanding in the context of mitochondrial alterations with aging in neurons stems from studies in the context of age-related neurodegenerative disease, including Parkinson’s disease (PD) and Alzheimer’s disease (AD) [28]. In view of the aforementioned similarity in involvement of neuromuscular junction degeneration, mitochondrial changes in motoneurons in ALS are also likely to have some relevance for aging. In contrast, there have been many studies characterizing multiple facets of mitochondrial structure [29] and function in aging muscle fibers [30,31,32]. However, at least some of these mitochondrial changes in aging muscle appear to occur selectively in the denervated muscle fibers that accumulate in advanced age [14,33], underscoring the difficulty in addressing mitochondrial changes in muscle fibers independently from changes occurring in motoneurons with aging. A summary of cited papers addressing mitochondrial structure and function changes with aging is provided in Table 1.

### 3.1. Mitochondrial Morphology

Our understanding of mitochondrial structure has evolved remarkably in recent years. Not only is the structure far more complex than the bean-shaped organelles that are still found depicted in some textbooks, but the structure is dynamic and tightly regulated. Perhaps nowhere else has this evolution in understanding of mitochondrial structure been more dramatic than in skeletal muscle. As early as 1978, the first indications that mitochondria likely formed a reticular-like structure in skeletal muscle were presented, based upon three-dimensional reconstruction of serial transverse electron micrograph images of rat diaphragm muscle [34]. This was later corroborated in human skeletal muscle using scanning electron microscopy of freeze-fractured muscle that had been depleted of the contractile elements to better expose the mitochondria and sarcoplasmic reticular structures [35]. Even within the crowded space of a muscle fiber, this structure is dynamic, with recent evidence showing increased mitochondrial fusion during an acute bout of exercise in human skeletal muscle [36]. The function of the connectivity of mitochondrial structure within skeletal muscle has long been speculated as perhaps facilitating ATP production in the deep regions of muscle fibers that are far away from the oxygen source of the microcirculation [37]. In this respect, the majority of mitochondrial connectivity appears to occur in the transverse/x-y orientation of a muscle fiber, where mitochondria weave around and between the myofibrils [38,39], and the mitochondrial protein composition is specialized to favor generation of the proton-motive force at regions close to capillaries and to favor ATP generation at regions distant from capillaries [40]. These are structural and biochemical arrangements that would indeed facilitate ATP generation within the deep regions of muscle fibers that are distant from a source of oxygen. Furthermore, recent evidence supports the idea that the mitochondrial reticular structure operates as a functional grid to propagate the proton-motive force [40,41]. To our knowledge, mitochondrial structure in motoneurons has not been deeply interrogated, but the smaller diameter of motoneurons and their axons means that diffusion of oxygen is less likely to be limiting to mitochondrial respiratory function, and thus, mitochondrial reticular structure is presumably not necessary to the same degree.

Mitochondria in aged skeletal muscle have been reported to be both larger and more complex [29], and smaller and less complex [42,43], than in young adulthood. However, major limitations to the analyses to date include: (i) no studies have used methods that permit assessment of the three-dimensional structure of mitochondria, although one study has partially addressed this by making measures in both longitudinal and transverse sections of muscle [29]; (ii) prior studies have not accounted for the fact that muscle is comprised of different fiber types that have distinct mitochondrial morphology [35,38], noting that fiber type can shift in opposing directions between muscles of contrasting fiber type dominance with aging [44]; and most importantly, (iii) no prior studies have addressed the heterogeneity in fiber-to-fiber affect in aging muscle, including the fact that denervated fibers are interspersed among normal fibers [7,23]. Whereas the technology to resolve the first issue is increasingly available (e.g., focused ion beam scanning electron microscopy), the latter two issues are related to the very small amount of tissue typically sampled in electron microscopy studies, coupled with a lack of correlative information about the muscle fibers being sampled (e.g., their fiber type and innervation status), and are more challenging to resolve. To illustrate the likely importance of the sampling issues noted in point (iii), the disparate findings of more highly fused [29] versus more fragmented mitochondria [42] between different studies in aging muscle could easily be affected by inadvertent sampling of denervated muscle fibers in the study reporting highly fragmented mitochondria [42], since denervation causes mitochondrial fragmentation [45,46]. In support of this possibility, most of the mice used in the study that reported more fused mitochondria [29] were not at as advanced an age or stage of muscle atrophy as the rats used in the study finding more fissioned mitochondria [42], and we have shown a large abundance of denervated muscle fibers [7] in rats of the same age as the study finding more fissioned mitochondria [42]. Clearly, further studies are required to address these issues.

To our knowledge, only one study has assessed mitochondrial morphology in aging motoneurons. This study found that mitochondria in motoneuron terminals at the neuromuscular junction in aged rats were swollen, had distorted cristae, and sometimes displayed evidence of membrane rupture [47] (Figure 3). Notably, these are all features of mitochondria that have undergone permeability transition, an event associated with release of pro-apoptotic proteins from the mitochondria consequent to the formation of a large non-specific pore across the inner mitochondrial membrane that causes mitochondrial swelling and subsequent rupture of the outer mitochondrial membrane. The likely causes of mitochondrial permeability transition with aging are considered in Section 4, below. In contrast to the motoneuron terminals, mitochondria in the motoneuron cell bodies of aged rats were not swollen and had intact cristae, although their morphology was still somewhat altered from that seen in young adult rats [47]. Therefore, the limited evidence available suggests that abnormal mitochondrial structure also occurs in motoneurons with aging.

### 3.2. Mitochondrial Function

Interrogating the functional consequences of alterations in mitochondrial structure is an evolving landscape, too. In the context of skeletal muscle, early studies making direct measures of mitochondrial function used mechanically isolated organelles [48]. However, we have shown that routine isolation procedures can profoundly affect both mitochondrial structure and measures of mitochondrial function [49]. In the context of aging, we showed that mechanical isolation of mitochondria not only exaggerated the degree of dysfunction, but some functional alterations seen with aging were unique to isolated mitochondria [31]. More recent studies frequently employ a technique involving saponin-permeabilization of muscle fibers [50], an approach that leaves mitochondrial network structure intact and allows all mitochondria within the tissue to be represented (isolation typically yields 50% of the mitochondria within a given tissue) [51]. As we have reviewed previously, the conclusion of these studies is that there is a reduction in mitochondrial respiratory capacity, a variable increase in mitochondrial ROS, and increased susceptibility to permeability transition in aging muscle [48,52]. Interestingly, similar to what has been reported for aged motoneurons [47] (Figure 3, above), not only are mitochondria in aging skeletal muscle sensitized to permeability transition, but, based upon an increased fraction of nuclei containing the mitochondrial-derived protein EndoG, some mitochondria in aged muscle fibers have undergone permeability transition [32] (Figure 4). Whether these changes in muscle mitochondria are causally involved in neuromuscular junction degeneration with aging is not clear. On the contrary, as noted above, we have evidence that the mitochondrial function changes seen in aged human muscle are at least partly secondary to the denervation of muscle fibers in aged muscle [14,33]. Nevertheless, there is evidence that experimentally induced mitochondrial impairment in muscle can adversely affect the neuromuscular junction, as shown in experiments involving muscle-specific over-expression of uncoupling protein 1 (*UCP1*). These experiments showed that muscle-specific *UCP1* over-expression induces alterations in neuromuscular junction morphology, including AChR fragmentation and degeneration of the motoneuron terminals [53]. However, the significance of this mechanism for aging skeletal muscle is unclear because *UCP2* and *UCP3* are the major UCP isoforms in skeletal muscle and their role in mitochondrial uncoupling is less clear [54,55]. Whether other mechanisms of mitochondrial impairment in muscle fibers can cause denervation remains a work in progress. Indeed, a significant limitation of current evidence is that we know nothing about the function of the mitochondria within muscle fibers specifically at the endplate. This is clearly an issue requiring additional study.

As noted above, mitochondria in motoneuron terminals of aged rats exhibit morphological features indicative of having undergone permeability transition [47]. Consistent with this interpretation, this same study observed mitochondria releasing cytochrome c in motoneuron terminals with resulting activation of caspase 3 (Figure 5), and a co-localization of activated caspase-3 with the retrograde transport motor protein dynein in motoneuron axons (Figure 6), suggesting retrograde transport of apoptotic factors to the motoneuron soma. These findings argue that with aging mitochondria undergoing permeability transition in the motoneuron terminals can cause degeneration of the motoneuron through retrograde signaling [47]. This is consistent with other observations in aging muscle suggesting a motoneuron die-back phenomenon [56]. Interestingly, this ‘dying-back’ phenomenon is a common feature in several neurodegenerative conditions affecting muscle [57,58,59], and in these settings mitochondria in the motoneuron terminals are frequently affected [59,60], supporting the notion that mitochondrial-mediated axonal die-back is likely relevant to aging muscle atrophy. A key question, however, is what factors are responsible for accumulation of dysfunctional mitochondria in the terminals.

## 4. Mechanisms of Mitochondrial Dysfunction with Aging

In looking to explain why alterations in mitochondria occur with aging, multiple facets of mitochondrial biology need to be considered, including the fact that they have their own genome, that their structure is dynamically regulated (fusion and fission), that they are transported long distances in motoneurons to the terminals at the neuromuscular junction, and that they undergo continual turnover involving degradation (mitophagy) and replacement (mitochondrial biogenesis) to ensure high fidelity of organelle function. As such, understanding the basis for mitochondrial alterations with aging requires consideration of how aging impacts mtDNA (mutation and copy number), regulation of mitochondrial dynamics, and the mechanisms governing mitochondrial quality control or mitostasis. A summary of cited papers addressing mechanisms that may contribute to impaired mitochondrial function in muscle versus neurons with aging is provided in Table 2.

### 4.1. mtDNA Alterations

Mitochondria contain their own genome, the 16.6 kp long mitochondrial DNA (mtDNA), which consists of 37 genes encoding 13 polypeptides of the respiratory chain complexes, 2 rRNAs and 22 tRNAs [61]. The remaining assembly factors and proteins required to generate functional mitochondria are encoded by the nuclear DNA (nDNA), underscoring the need for highly coordinated expression of genes encoded by nDNA and mtDNA during mitochondrial biogenesis. In addition, mtDNA is strictly maternally inherited, and it is vulnerable to mutation. Indeed, there is an abundance of evidence for a progressive accumulation of mtDNA mutations in skeletal muscle with aging [62,63,64]. Although a lack of histone proteins, poor mtDNA repair mechanisms, and the close proximity of mtDNA to sites of ROS production have all been suggested as reasons for mtDNA accumulating damage with aging, each of these factors have largely been disproven [65] and the search for the causes of mtDNA mutation in aging and disease remains an important area of research [61].

A special feature of mitochondria is the presence of multiple genomes per mitochondrion, such that when a mutation is present, both mutant and wild-type mtDNA copies co-exist in a single cell, in multiple cells of a given tissue, and/or in different tissues of a given individual. This phenomenon is known as heteroplasmy and it dictates whether normal mitochondrial function can be maintained [61]. In fact, mutant mtDNA copies must reach a critical threshold (60–90% heteroplasmy) within a given cell before mitochondrial function begins to be compromised, because at this threshold new synthesis of mitochondria is now largely being made from damaged templates [66,67]. In the case of mtDNA depletion, the concept of a critical level to be reached before a biochemical defect occurs may also be applicable. For example, in the context of mitochondrial disease, pre-existing mtDNA depletion will reduce the absolute number of mtDNA deletions needed before the critical threshold causing mitochondrial functional impairment is reached [68]. A similar situation exists in the age- and smoking-related disease, chronic obstructive pulmonary disease, where locomotor muscle from patients exhibits both an exacerbation of mtDNA mutation accumulation and mtDNA depletion [69]. It is generally thought that the consequences of high levels of mtDNA mutations are most relevant in cells that have high reliance upon mitochondria for energy metabolism, such as neurons and muscle fibers [70]. While there is no doubt that mtDNA alterations have devastating consequences on mitochondrial and cellular function when they reach critical threshold, a major limitation to date is a lack of clear evidence that the burden of these mtDNA alterations are sufficient to make a meaningful contribution to aging muscle. In this respect, whereas the role of mtDNA depletion as a mechanism of mitochondrial impairment in aging skeletal muscle has not been widely studied, much has been made of the accumulation of mtDNA mutations with aging in skeletal muscle. Studies have shown that mtDNA deletions accumulate with aging in a segmental manner along the length of individual muscle fibers and fibers become mosaic containing a mixture of normal and dysfunctional mitochondria [71]. One of the first studies to directly link mtDNA mutations to aging muscle atrophy showed in aged rats that on occasion there is a co-localization of cytochrome oxidase deficient myofiber segments harboring high levels of mtDNA mutations, with fiber atrophy and breakage [72]. The prevailing hypothesis put forward is that dysfunctional mitochondria expand along the length of the muscle fiber, resulting in impairment of normal cellular homeostasis, increased oxidative damage, and activation of apoptotic and necrotic cell death pathways that precipitates atrophy and loss of the muscle fiber [73]. It was also suggested that the respiratory chain deficiency in the muscle segment could result in neuromuscular junction degeneration [72], although no experiments were done to test this possibility. Despite the logical appeal of a mtDNA mutation mechanism of muscle atrophy, the role of mtDNA alterations in driving muscle atrophy appears to be trivial compared to other processes causing atrophy. Specifically, in a study of aged human limb muscle, it was reported that 95% of muscle fiber segments that harbor high levels of mtDNA mutations and severe mitochondrial oxidative impairment of the mitochondria do not exhibit atrophy [74]. Similarly, data from our group show that cytochrome oxidase deficient muscle fibers are not among the smallest muscle fibers in very old rodent muscle [8] and occur at a frequency (< 1%) that is far lower than the abundance of severely atrophied denervated muscle fibers [7]. Furthermore, as has been pointed out previously, patients with primary mitochondrial disease have much higher burdens of mtDNA mutation than seen with normal aging, yet their primary muscle phenotype is one of severe exercise intolerance and weakness rather than atrophy [75,76]. On this basis, it may be time to retire the idea that mtDNA mutation accumulation in muscle is causally related to atrophy.

In contrast to the relatively uninspiring data from naturally occurring aging, a mouse model exhibiting a mutation in the proof-reading activity of polymerase gamma (*PolG*) exhibits spontaneous mtDNA deletions and depletion, leading to an early appearance of traits that resemble premature aging [77], including muscle atrophy [78]. With respect to mitochondrial phenotype, these mice exhibit lower levels of complex I, III and IV compared to wildtype, accompanied by reduced maximal respiratory capacity [78]. However, this is not consistent with aging, where muscles with similar fiber type to that examined in the *PolG* mutant mouse study (fast twitch dominant) exhibit higher (not lower) levels of protein constituents of complex I, III and IV in very advanced age when atrophy becomes severe [79]. Thus, evidence from *PolG* mutant mice suggest that high levels of mtDNA mutations and/or mtDNA depletion lead to different mitochondrial phenotypes than occur with normal aging at ages associated with severe muscle atrophy. This speaks to the fact that no prior studies have directly compared the nature of the muscle atrophy seen with normal aging to that occurring in the *PolG* mutant mouse to determine if mtDNA mutations and/or depletion phenocopies known aging muscle attributes. Such an analysis should include determining whether the *PolG* mutant mouse also exhibits neuromuscular junction degeneration, motor unit remodeling, and a failed reinnervation phenotype as seen with aging. These are areas that warrant further investigation to more fully understand the potential for mtDNA alterations to contribute to aging muscle deterioration.

With regard to aging neurons, the impact of mtDNA defects (depletion and mutations) has been studied in the context of age-related diseases such as PD; however, the role of mtDNA in motoneurons in the context of aging muscle atrophy remains under-studied. To our knowledge, there is only one study relevant to this point, and it was performed in post-mortem samples of spinal cord motoneurons of 14 elderly individuals (68–99 years of age) [80]. This study revealed mtDNA depletion, but not significant levels of mtDNA mutations, and this was associated with motoneuron loss, reduction in the motoneuron size, and complex I respiratory chain deficiency [80]. In addition, only one COX-deficient motoneuron was detected using COX/SDH histochemistry. This latter observation suggests either that mtDNA mutations make minimal contribution to impaired mitochondrial function in aged neurons, or perhaps that neurons with COX deficiency die so quickly that there is little opportunity for their detection. Regardless, these findings suggest that mtDNA depletion may be an important mechanism of impaired motoneuron respiratory function with aging. Clearly, further work is needed.

### 4.2. Mitochondrial Dynamics and Mitostasis

Mitochondrial dynamics govern mitochondrial number, size, morphology and function through the exchange of mitochondrial content (mtDNA, proteins and metabolites) to meet variable energy demands, facilitate mitochondrial quality control, and spatially distribute mitochondria within a cell [81]. This is dictated by the tight balance between two opposing processes: mitochondrial fusion and fission. Fusion is catalyzed by GTPases that are embedded in the inner and outer mitochondrial membranes and involves two reactions: (i) fusion of the outer mitochondrial membrane, which is regulated by mitofusins 1 and 2 (*Mfn1* and *Mfn2*); and (ii) fusion of the inner mitochondrial membrane by optic atrophy 1 (*Opa1*). On the opposing side, mitochondrial fission is the process whereby a mitochondrion divides into two daughter mitochondria. Fission occurs under conditions of elevated stress and cell death, resulting in mitochondrial network fragmentation, a morphological state associated with mitochondrial dysfunction [82]. Portions of dysfunctional mitochondria (e.g., having low membrane potential) may also be fissioned off and degraded as part of the mitostatic maintenance of mitochondrial function [81] (see below). In mammals, fission is principally caused by dynamin-related protein 1 (*Drp1*), which forms a constrictive ‘necklace’ around regions of the mitochondrial network that tightens until two daughter mitochondria are formed. Fission also involves mitochondrial fission factor (*Mff*) and fission protein 1 (*Fis1*), which recruit *Drp1* in the mitochondrial membrane [83]. Importantly, disruptions in either fusion or fission processes (e.g., by mutations in mitochondrial dynamics proteins) each lead to muscle pathology (Figure 7). For example, single knockout of the pro-fusion proteins *Opa1* [84] or *Mfn2* [85] impairs mitochondrial function and leads to early muscle atrophy. Deletion of fission-inducing Drp1 in flies depletes mitochondria from the motoneuron terminals, which causes impaired neuromuscular transmission during nerve stimulation [86]. Muscle-specific knockout of the fission protein *Drp1* causes premature lethality (30 d postnatal) associated with muscle fiber loss, muscle fiber atrophy, and a more fused mitochondrial network [87]. Furthermore, acute double-knockout of the fusion protein *Opa1* and the fission protein *Drp1* leads to various indices of abnormal mitochondrial morphology (e.g., disrupted cristae, hyperfused and enlarged mitochondria) and inhibits removal of dysfunctional mitochondria by mitophagy, which is accompanied by muscle atrophy and weakness [88]. Interestingly, muscle-specific *Opa1*/*Drp1* double knockout mice also exhibited indices of muscle denervation that resolved over time, suggesting that impaired mitochondrial dynamics in muscle alone can induce neuromuscular junction instability [88]. The importance of mitochondrial dynamics in motoneurons is underscored by the fact that some genetic mutations of mitochondrial dynamics regulatory genes are associated with neuromuscular disorders. For example, mutations in *Mfn2* cause Charcot-Marie-Tooth type 2A (CMT2A), an autosomal dominant disorder characterized by loss of motoneurons resulting in muscle atrophy [89]. Therefore, impaired mitochondrial dynamics has severe consequences for muscle and motoneurons, supporting the idea that the observed reduced expression of both the fusion and fission proteins in aging muscle [84] are contributing to the mitochondrial impairment and muscle atrophy observed with aging.

Mitochondrial dynamics are directly linked to mitostasis, which is the preservation of mitochondrial health over time. As recently reviewed by Misgeld and Schwarz [90], mitostasis is achieved by a highly efficient mitochondrial quality control system comprised of three main components [83]. Firstly, individual mitochondrial proteins that are misfolded and/or oxidized can be removed by the AAA+ proteases, which are found in the matrix and embedded in the mitochondrial inner membrane. Secondly, larger assemblies of damaged mitochondrial proteins and lipids can bud off individual mitochondria into mitochondria-derived vesicles (MDVs) and be targeted for degradation by lysosomes and other pathways. Finally, bulk degradation of a whole dysfunctional mitochondrion occurs by mitophagy, wherein dysfunctional organelles are targeted by specific proteins for lysosomal degradation. This provides for a highly efficient graduated mitochondrial quality control system, to permit removal and replacement of individual proteins and membrane all the way to complete organelle replacement. Although these systems operate with high fidelity most of the time, they can falter and should thus be considered in understanding the reasons for mitochondrial impairment with aging.

The importance of selective replacement of individual mitochondrial proteins is underscored by the remarkable heterogeneity in half-lives of different mitochondrial proteins [91]. Although AAA+ proteases have not yet been studied in the context of aging muscle atrophy, mutations of *m*-AAA proteases (one class of AAA+ proteases) are associated with neurodegeneration, and deletion of *m*-AAA protease increases susceptibility to mitochondrial permeability transition [92]. Thus, alterations in *m*-AAA proteases could play a role in the accumulation of mitochondria having undergone permeability transition in the motoneuron terminals of aged rats [47]. Similarly, loss of the AAA+ protease *YME1L*, which not only degrades damaged or non-assembled proteins in the mitochondrial inner membrane but also cleaves Opa1 to activate it and catalyze inner membrane fusion [93], causes ocular degeneration and axonal degeneration in the spinal cord [94]. The potential role of alterations in function of AAA+ proteases in muscle and motoneuron mitochondrial impairment with aging is unknown, but is worthy of consideration based upon the aforementioned effects.

As noted above, mitophagy refers to the bulk degradation of dysfunctional mitochondria. The classical view of mitophagy involves a signaling pathway that includes the serine/threonine *Pink1* and the E3 ubiquitin ligase, *Parkin*. *Pink1* is transcribed in the nucleus, translated in the cytosol, and imported into the mitochondria through transporter of outer membrane (*TOM*) and transporter of inner membrane (*TIM*) complexes, ultimately localizing on the outer mitochondrial membrane of mitochondria. Although *Pink1* is normally unstable due to high degradation rates, when mitochondria lose their membrane potential, Pink1 stabilizes on the OMM and initiates mitophagy by recruiting *Parkin*. Whereas the importance of both *Pink1* and *Parkin* to mitophagy is well-established in neurons in general [90,95], and thus, is presumably relevant in motoneurons too, recent evidence suggests mitophagy occurs in the absence of *Pink1* stabilization in skeletal muscle [96]. This nuance notwithstanding, double knockout of *Pink1* and *Parkin* causes motoneuron axon swelling and post-synaptic AChR cluster fragmentation at the neuromuscular junction degeneration [60]; changes that are similar to what occurs with aging. Furthermore, Parkin knockout impairs some aspects of mitophagy in skeletal muscle [97], sensitizes mitochondria in muscle to permeability transition [98] and causes muscle atrophy [99]. It is also interesting to note that patients with PD, a disease wherein impairments in mitophagy are well-established, exhibit an exacerbation of age-related motor unit remodeling [100]. Finally, recent evidence suggests that whereas mitophagy can occur independently of *Parkin* in neurons [101], knockout of *Pink1* or *Parkin* prevents the age-related increase in mitophagy and exacerbates mitochondrial dysfunction [102]. Thus, impaired mitophagy is very likely to play an important role in the accumulation of dysfunctional mitochondria with aging, and, in turn, contribute to neuromuscular junction instability that underlies denervation in aging muscle. Consistent with this possibility, we [32] (Figure 8) have observed reduced levels of *Parkin* protein in aged human limb muscle. As alluded to above, in addition to its role in mitophagy, *Parkin*, ubiquitinates mitofusins and other mitochondrial proteins, and is thus involved in regulating mitochondrial network fragmentation and autophagic turnover of mitochondrial proteins [103]. Parkin is also involved in formation of MDVs [104]. As such, the above points underscore the interconnection between mitochondrial dynamics and mitostatic pathways.

### 4.3. The Special Problem of Mitostasis in Motoneurons

As stated above, biosynthesis of mitochondria requires the coordinated expression of both nuclear DNA- and mtDNA-encoded genes. The classical view is that mitochondrial biogenesis must, therefore, occur in close proximity to a nucleus. Whereas the multi-nucleated nature of skeletal muscle fibers provides for recurring so-called ‘nuclear domains’ along their length [105], motoneurons (and other neurons) are mononucleated, which creates a significant challenge for synthesis of new mitochondria for the motoneuron terminals that are located a long distance from the nucleus-containing soma [90]. Again, the traditional view has been that mitochondria are synthesized in the soma and then transported down the axon to the most distant axon terminals. Although there are motor proteins along the microtubule and other elements of the cytoskeleton that participate in axonal transport in motoneurons, and axonal transport is reported to be slower in aged rats and correlates with the degree of muscle atrophy [106], the role of mitochondrial transport as a key mechanism for replenishing mitochondria in motoneuron terminals is questionable. Specifically, the idea that mitochondrial transport is essential for maintenance of neuronal function has recently been challenged mainly due to the fact that the distance covered by a mitochondrion from the neuron soma to the synapse can exceed the lifetime of mitochondrial proteins, implicating other mechanisms in maintaining mitochondria in the portions of neurons that are distant from the soma [90]. In support of this idea, Parkin is recruited to damaged mitochondria in neuronal axons and leads to formation of autophagosomes [107], showing that mitophagy can occur locally, rather than requiring transport back to the neuron soma. Furthermore, there is emerging evidence that synthesis of mitochondrial proteins [108] and replication of mtDNA can occur in the axons [109]. It has also been suggested that new mitochondria may be supplied to axons by the transfer of the mitochondria from surrounding astrocytes [110]. Whether the perisynaptic Schwann cells that envelop the motoneuron terminals at the neuromuscular junction have the same capability should be determined, as there is already evidence that degenerating motoneuron terminals release factors that activate perisynaptic Schwann cells [111,112]. Therefore, these new findings highlight new areas for exploration to understand the interplay between mitochondrial dynamics, quality control and biogenesis in the motoneuron and how this may become compromised with aging.

## 5. Conclusions

There is strong evidence for alterations in mitochondria with aging in both muscle fibers and motoneurons. Whereas there is experimental evidence that muscle-specific impairment of mitochondria can cause neuromuscular junction degeneration, it is not yet clear whether this occurs with aging. Indeed, there is evidence that some of the mitochondrial alterations seen in very advanced age in muscle are a consequence rather than a cause of denervation of muscle fibers. On the other hand, there is evidence that mitochondria in the terminals of aged motoneurons have undergone permeability transition and that this likely causes motoneuron die-back through retrograde transport of pro-apoptotic proteins to the motoneuron soma. As summarized in Figure 9, in considering the potential mechanisms causing mitochondrial impairment with aging, whereas there is not strong support for mtDNA alterations in skeletal muscle fibers being involved in their atrophy with aging, there is some evidence that mtDNA depletion in aged human motoneurons may contribute to motoneuron dysfunction and death. Furthermore, although there is evidence for impaired mitochondrial dynamics in skeletal muscle, whether these changes are a cause or consequence of denervation remains unknown. Among the most likely causes of increased mitochondrial permeability transition in aging motoneurons and muscle fibers based on the evidence so far, is an impairment in mitostasis that specifically involves impaired Parkin-mediated mitophagy. However many important questions remain, including: (i) how function is affected in mitochondria immediately beneath the neuromuscular junction in muscle fibers and whether this is involved in denervation; (ii) whether perisynaptic Schwann cells contribute mitochondria as part of mitostasis in motoneuron terminals and whether this is adversely affected with aging; and (iii) whether mitochondria are involved in mediating the superior reinnervation capacity that is seen in populations with better retention of muscle mass and function in advanced age.

## Figures and Tables

**Figure 1 cells-09-00197-f001:**
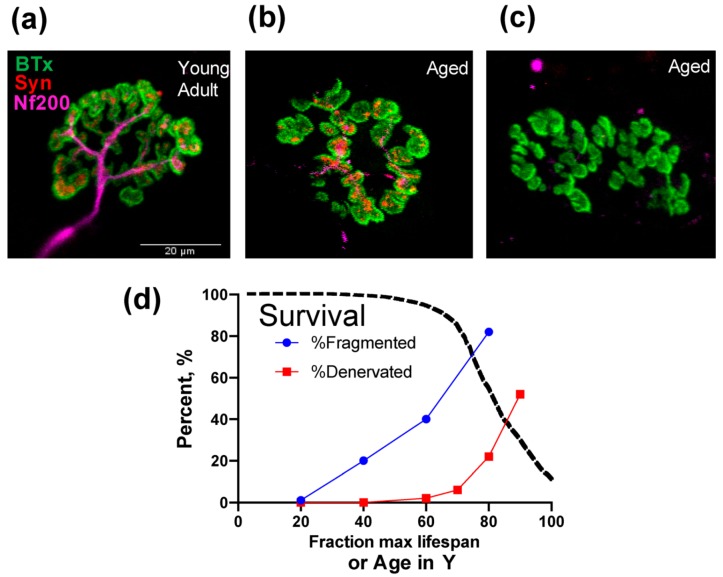
Impact of aging on the neuromuscular junction. Immunofluorescently labeled images of the neuromuscular junction from Young Adult (**a**) and Aged (**b**,**c**) Fisher 344 x Brown Norway rat gastrocnemius muscle reveal fragmentation of the post-synaptic AChR clusters (**b**) and some endplates that completely lack detectable motoneuron terminals (**c**). BTx = α-bungarotoxin to label the post-synaptic AChRs; Syn = synaptophysin to label the motoneuron terminals; Nf200 = neurofilament 200 to label the motoneuron axons. Panel (**d**) is based upon composite information about neuromuscular junction fragmentation and denervation from mouse [19], and muscle fiber denervation from rat [7] and human [14] skeletal muscle, to depict the time-course of neuromuscular junction fragmentation (blue line) and muscle fiber denervation (red line) relative to maximal lifespan and species-specific survival characteristics (dashed line).

**Figure 2 cells-09-00197-f002:**
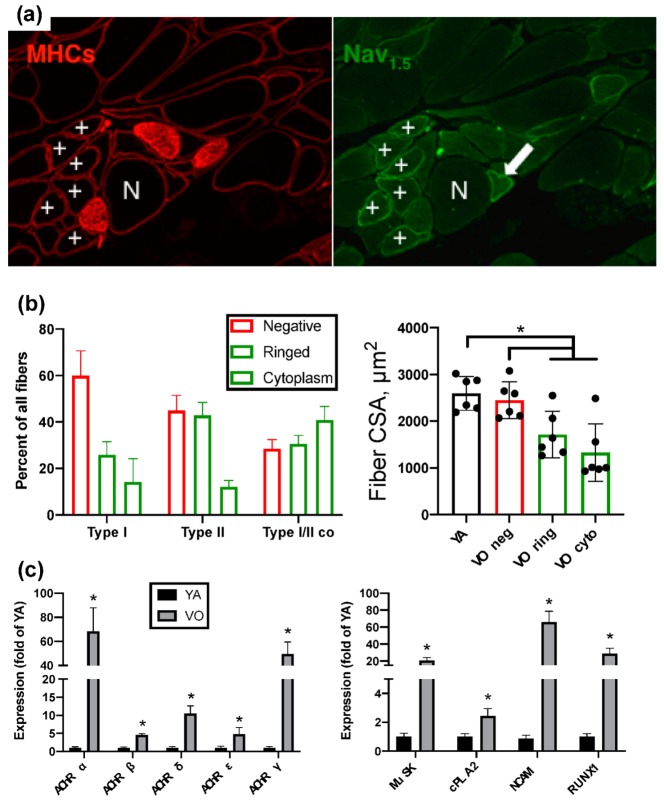
There is a large accumulation of denervated muscle fibers in advanced age, based upon the expression of the denervation-responsive sodium channel isoform, Nav_1.5_, in either the sarcolemmal region (ring; +) or cytoplasm (arrow). (**a**) Left panel shows muscle fibers from 35 mo old rat gastrocnemius muscle labeled with primary antibody against MHCs = slow (type I) myosin heavy chain and laminin (both on same fluorescent secondary antibody). Right panel shows a serial section where muscle fibers are labeled with primary antibody against Nav_1.5_. N = Nav_1.5_ negative fiber; “+” symbols identify Nav_1.5_ positive fibers. (**b**) Nav_1.5_ status by fiber type, and size of fibers by Nav_1.5_ status. Filled circles in right hand panel represent the individual values for each animal. (**c**) The expression of AChR subunits are elevated in very old muscle, particularly the reinnervation-denervation responsive α and γ subunits, coincident with a marked upregulation of other denervation-responsive genes (*MuSK, cPLA2, NCAM, RUNX1*). YA = young adult; VO = very old (35 mo old). * *p* < 0.05 versus VO Nav_1.5_ ring and Nav_1.5_ cyto. Data are taken from [7,24].

**Figure 3 cells-09-00197-f003:**
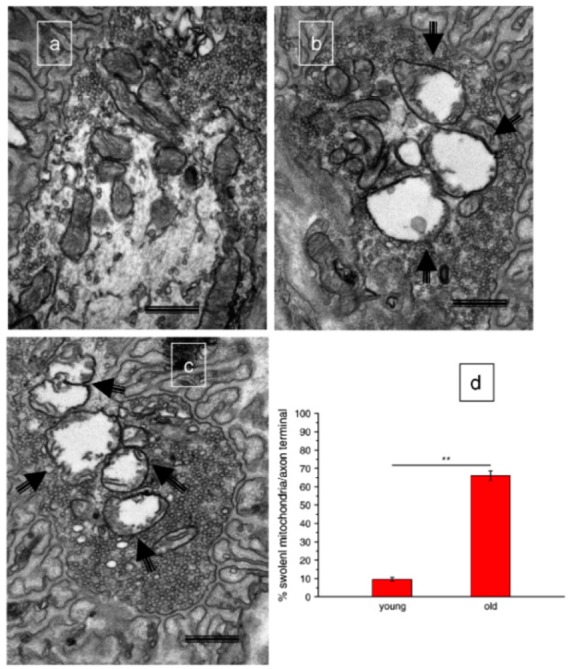
Compared to young animals (panel (**a**)) mitochondria in the motoneuron terminals of aged rats exhibit swelling, distorted cristae and membrane rupture (arrows in panels (**b**–**c**)); features that are consistent with having undergone mitochondrial permeability transition. (panel (**d**)) provides the percentage of swollen mitochondria in aged versus young rats. Bar = 500 nm. ** *p* < 0.001. Figure reproduced with permission from [47].

**Figure 4 cells-09-00197-f004:**
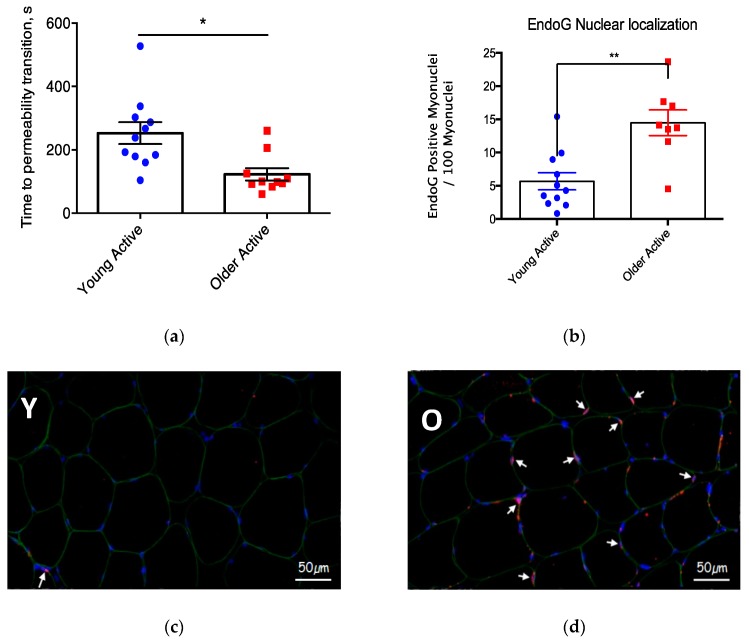
Mitochondria in aging skeletal muscle exhibit a sensitization to permeability transition in response to a Ca^2+^ challenge (represented by a shorter time to permeability transition; panel (**a**)), and this is associated with nuclear translocation of the mitochondrial-derived protein, EndoG (panel (**b**)). Filled blue circles = young physically active adult; filled red square = older physically active adult. Images show a human vastus lateralis muscle cross-section from a young physically active (Y; panel (**c**)) and an older physically active (O; panel (**d**)) man immunofluorescently labeled with primary antibodies against dystrophin (green), DAPI (blue), and EndoG (red). Arrows depict nuclei containing EndoG. * and ** *p* < 0.05 versus Young Active. Figures shown are adapted from [32].

**Figure 5 cells-09-00197-f005:**
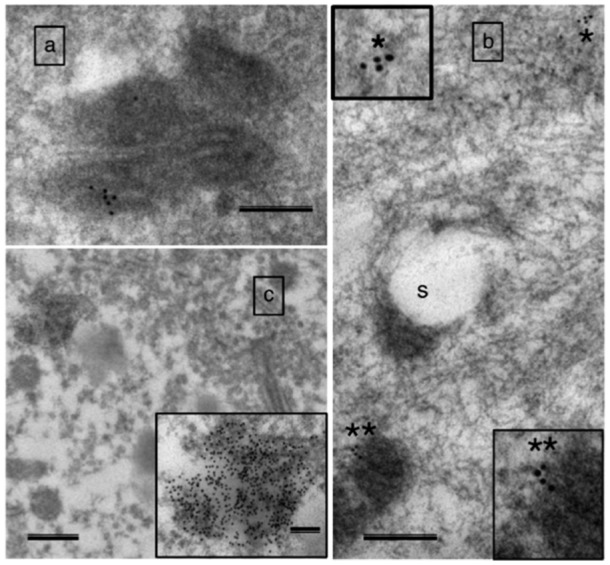
Some mitochondria in motoneuron terminals have released cytochrome c, associated with a corresponding activation of caspase 3. Panel (**a**) shows immunogold-labeled cytochrome c (black dots) located within the mitochondria in motoneuron terminals of a young rat, whereas in in old rat panel (**b**) cytochrome c is found in the cytoplasm (*) and there is evidence of cytochrome c being released from a mitochondrion (**). Panel (**c**) shows immunogold-labeled caspase 3 in the axonal cytoplasm. Figure reproduced from [47] with permission.

**Figure 6 cells-09-00197-f006:**
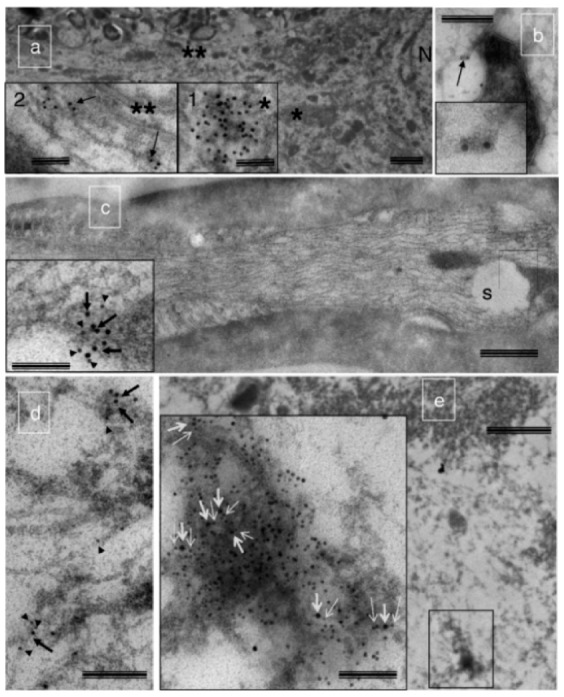
Spinal cord motoneuron soma have activated caspase 3, and activated caspase 3 is found to be associated with the axonal microtubules and the retrograde transport protein dynein, suggesting retrograde transport of activated caspase 3 from motoneuron terminals occurs with aging. Panel (**a**) shows immunogold-labeled activated caspase 3 in close proximity to the nucleus of the motoneuron soma, and associated with the microtubules in the axon hillock in aged rat. Panel (**b**) shows activated caspase 3 on vesicle-like structures associated with the microtubules in motoneuron axon. Panel (**c**) shows double immunogold labeling of activated caspase c (10 nm gold particle, arrows) and dynein (5 nm gold particle, arrowheads) in the proximity of a swollen mitochondrion. Panel (**d**) shows colocalization of dynein (arrowheads) and activated caspase 3 (arrows) with axon terminal microtubules. Panel (**e**) shows the soma of a motoneuron from an aged rat with double immunogold labeling of activated caspase 3 and dynein. Inset shows a magnified view co-localized dynein (thin arrows) with activated caspase 3 (thick arrows). * = caspase 3 immunoreactive spots; ** = caspase 3 associated with microtubules in the axon hillock. Figure reproduced from [47] with permission.

**Figure 7 cells-09-00197-f007:**
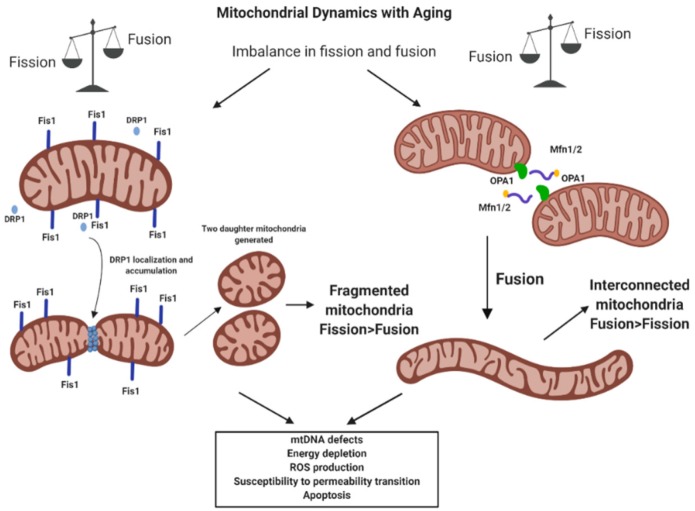
Schematic depicting the consequences of alterations in mitochondrial dynamics with aging.

**Figure 8 cells-09-00197-f008:**
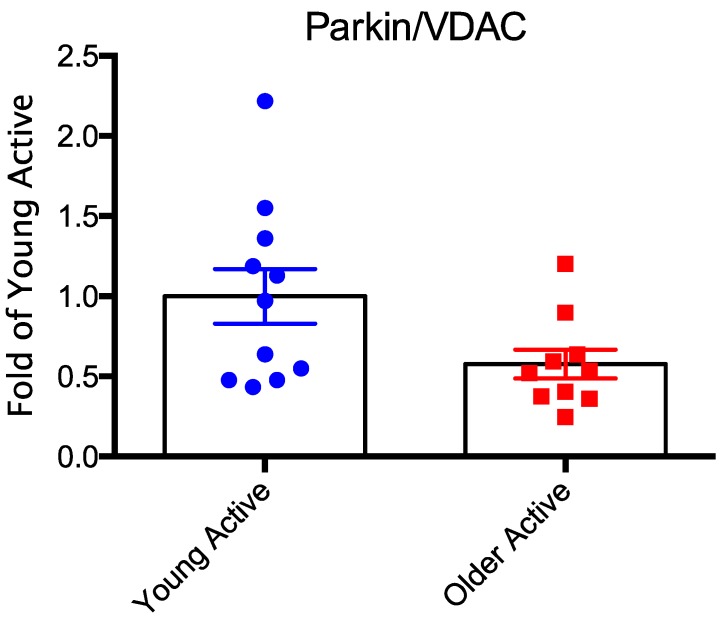
Parkin protein relative to a marker of mitochondrial content (voltage dependent anion channel; VDAC) is reduced in skeletal muscle of older physically active men. Filled blue circles = young physically active; filled red squares = older physically active. Figure adapted from [32].

**Figure 9 cells-09-00197-f009:**
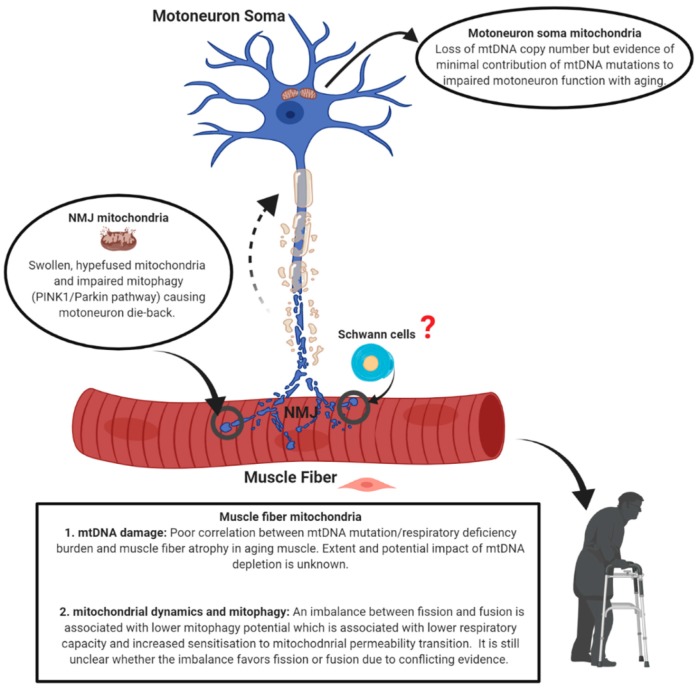
Schematic summarizing the global changes in mitochondria within skeletal muscle fibers and motoneurons with aging.

**Table 1 cells-09-00197-t001:** Mitochondrial structure and function changes with aging.

Aspect	Tissue	References
Mitochondrial structure	Muscle	[29,42,43]
Mitochondrial structure	Neuron	[47]
Mitochondrial function	Muscle	[14,30,31,32,33,79]
Mitochondrial function	Neuron	None

**Table 2 cells-09-00197-t002:** Mechanisms of impaired mitochondrial function with aging.

Mechanism	Tissue	References
mtDNA alteration	Muscle	[62,63,64,72,73,74,78]
mtDNA alteration	Neuron	[77,80]
Mitochondrial dynamics	Muscle	[84,85,87,88]
Mitochondrial dynamics	Neuron	[86,89]
Mitostasis	Muscle	[97,98,99]
Mitostasis	Neuron	[60,94,100,102,103,107,109,110]

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
