# Peer review of "Mitochondrial Mechanisms of Neuromuscular Junction Degeneration with Aging"

_cells, 2020, doi:10.3390/cells9010197_

Round 1

Reviewer 1 Report

This manuscript provides an up-to-date view of the role of mitochondria in: 1) aging and degenerating neuromuscular junctions (NMJs), and 2) age-related declines of skeletal muscle mass and strength (sarcopenia). While this topic (aging and mitochondria) is not particularly novel, the approach used by the authors to address it here is. To the point, they not only look at the "dying back" effects of aging of mitochondria on NMJs and thus muscle, but also the effects of age-related degeneration of mitochondria in the pre-synaptic nerve terminals of NMJs. Indeed, this is the novel take on the effects of mitochondrial damage to NMJ stability, and by extension, maintenance of muscle mass and neuromuscular function during the natural process of aging. The authors are to be commended for this thorough, yet succinct review of the literature on this meaningful, yet understudied topic (aging of mitochondria, especially in neurons). There is little the reviewer can suggest except that the manuscript might benefit from the inclusion of 2-3 tables. This would not be to seek conformity with other reviews, but to provide easy to refer to summaries of findings concerning specific topics, e.g. effects of aging on sarcolemmal mitochondria, effects of aging on neural mitochondria, etc. Also, a figure showing fragmentation of endplate as discussed on p.3 is warranted.

Author Response

We thank you for your positive feedback. We have now included two new tables to summarize relevant references for section 3 (Mitochondrial Structure & Function) and section 4 (Mechanisms of Impaired Mitochondrial Function) and hope the reviewer finds these address the request.

Reviewer 2 Report

This is an interesting and valuable review that delves deeply into the possible mechanisms by which synaptic mitochondria impairment might occur in aging muscle and motor neurons, and how this might lead to muscle denervation, sarcopenia and weakness.

The evidence for a substantial contribution of denervation to sarcopenia needs to be better explained. The central theme of this review is the hypothesis that muscle denervation is a major cause of sarcopenia and that impaired mitochondrial function causes the muscle denervation. Indeed the abstract asserts that: “Denervation is a major driver of changes in aging muscle”. The latter assumption is so central to the hypothesis that it needs to be established for the reader by presenting the quantitative evidence, rather than just some citations. Section 2 is titled “Neuromuscular Junction, Denervation-reinnervation and Atrophy of Aging Muscle” but does not define the time-course of denervation in either humans or in any animal model. There are now a number of papers that present evidence for the time-course of muscle denervation in aging mice and also in mouse models of ALS. The authors should start this review by grounding it on the published quantitative evidence for the time-course of denervation in mice (c.f. sarcopenia). Why not replace the vague hypothetical graph in Fig 1 with actual data for at least on species (mice or rats) with the life-span on the x-axis and some measure of % muscle fiber denervation on the y-axis? Without real numbers, graphs don’t make much sense. If there is a problem with differing findings between species or conflicting results among studies then tell the reader what these problems are. One of the problems in reconciling the literature on age-related impairment of the NMJ is that the timepoint/s described as “aged” often vary substantially from study to study. It would be very helpful to present the degree of denervation with age (in months) and then discuss the degree to which impairments of mitochondria correlate with the time-course of denervation and muscle atrophy.

While fairly clear for the most part, the text contains some very long and convoluted sentences that need revision to improve clarity and a number of other corrections are needed (see below).

Specific points

Fig 1 is a cartoon graph that aims to explain the relationship between loss of reinnervation capacity of the motor neurons and the associated loss of muscle mass with age. The graph is poorly labelled and seems to have been distorted in the pdf. I did not understand what it is trying to convey, apart from the idea that sarcopenia occurs after loss of reinnervation capacity. “Reinnervation capacity” needs to be defined in terms of how it has been measured. What does “Muscle mass compatible with” mean? It would be more valuable to provide the reader with real quantitative data from the literature. Alternatively the graph should be labelled “a hypothetical model” with explanation in the legend.

Line (l.) 55-57 Consider breaking this sentence into two to improve clarity and ease of reading.

60-63 ditto. Try to keep sentences short each making a single point.

l.64-65 Define what you mean by denervation events. Do you mean: 1/ the denervation of individual motor endplate? Or 2/ the retraction of a whole motor unit? (i.e. coordinated at the level of the single motor neuron) or 3/ denervation of a whole muscle?

Fig 1: X-axis label “Ag” “Age”?

78 See also: N. Arizono, O. Koreto, Y. Iwai, T. Hidaka, O. Takeoka, Morphometric analysis of human neuromuscular junctions in different ages, Acta Pathol Japon 34 (1984) 1243-1249.

153 “Mitochondria in [aged?] skeletal muscle have…”

Early in the manuscript it would be helpful for readers who do not work on mitochondria to define what is meant by “permeability transition” and give an idea of how this is usually detected/measured.

210-217 Fragmentation of the endplate AChR plaque is not a measure of NMJ impairment. It can result from muscle fibre regeneration as shown by the late Wesley Thompson and does not necessarily imply any impairment of NMJ function as Clarke Slater has shown (cited earlier in the manuscript). Lets not perpetuate this discredited marker. The Dupuis paper presents better evidence of NMJ impairment.

226 “…terminals [that?] can cause…”

l.341-345 Please break this. difficult to read, sentence up. I can’t follow the meaning.

Author Response

This is an interesting and valuable review that delves deeply into the possible mechanisms by which synaptic mitochondria impairment might occur in aging muscle and motor neurons, and how this might lead to muscle denervation, sarcopenia and weakness.

The evidence for a substantial contribution of denervation to sarcopenia needs to be better explained. The central theme of this review is the hypothesis that muscle denervation is a major cause of sarcopenia and that impaired mitochondrial function causes the muscle denervation. Indeed the abstract asserts that: “Denervation is a major driver of changes in aging muscle”. The latter assumption is so central to the hypothesis that it needs to be established for the reader by presenting the quantitative evidence, rather than just some citations.

Thank you for raising this point. We have now revised what was Figure 2 in the original version to provide quantitative information establishing denervation as an important influencing factor in aging muscle.

 Section 2 is titled “Neuromuscular Junction, Denervation-reinnervation and Atrophy of Aging Muscle” but does not define the time-course of denervation in either humans or in any animal model. There are now a number of papers that present evidence for the time-course of muscle denervation in aging mice and also in mouse models of ALS. The authors should start this review by grounding it on the published quantitative evidence for the time-course of denervation in mice (c.f. sarcopenia).

Thank you for this suggestion. We have replaced the original Figure 1 with one that provides images of key neuromuscular junction structures and their modification with aging. This figure also provides a composite graphic that depicts the accumulation of muscle fibers with fragmented neuromuscular junctions and the accumulation of denervated muscle fibers. This latter graphic takes the time-course data for neuromuscular junction fragmentation and denervation (lack of motoneuron at endplate) from Valdez et al. 2010, and also incorporates data describing the accumulation of denervated muscle fibers from aged rat (Rowan et al. 2012) and humans (Spendiff et al. 2016), to depict the evolution of neuromuscular junction degeneration and denervation across the lifespan. To facilitate comparisons across the lifespan, we provide reference points to the fraction of maximal lifespan and the survival curve. We hope this is deemed valuable and a suitable response to the request made by the reviewer for quantitative information.

 Why not replace the vague hypothetical graph in Fig 1 with actual data for at least on species (mice or rats) with the life-span on the x-axis and some measure of % muscle fiber denervation on the y-axis? Without real numbers, graphs don’t make much sense. If there is a problem with differing findings between species or conflicting results among studies then tell the reader what these problems are. One of the problems in reconciling the literature on age-related impairment of the NMJ is that the timepoint/s described as “aged” often vary substantially from study to study. It would be very helpful to present the degree of denervation with age (in months) and then discuss the degree to which impairments of mitochondria correlate with the time-course of denervation and muscle atrophy.

Please see response above.

While fairly clear for the most part, the text contains some very long and convoluted sentences that need revision to improve clarity and a number of other corrections are needed (see below).

Specific points

Fig 1 is a cartoon graph that aims to explain the relationship between loss of reinnervation capacity of the motor neurons and the associated loss of muscle mass with age. The graph is poorly labelled and seems to have been distorted in the pdf. I did not understand what it is trying to convey, apart from the idea that sarcopenia occurs after loss of reinnervation capacity. “Reinnervation capacity” needs to be defined in terms of how it has been measured. What does “Muscle mass compatible with” mean? It would be more valuable to provide the reader with real quantitative data from the literature. Alternatively the graph should be labelled “a hypothetical model” with explanation in the legend.

Figure revised as noted above.

Line (l.) 55-57 Consider breaking this sentence into two to improve clarity and ease of reading.

Done as requested.

60-63 ditto. Try to keep sentences short each making a single point. Done.

l.64-65 Define what you mean by denervation events. Do you mean: 1/ the denervation of individual motor endplate? Or 2/ the retraction of a whole motor unit? (i.e. coordinated at the level of the single motor neuron) or 3/ denervation of a whole muscle?

This has now been defined in the text as denervation of an individual endplate. It may involve the death of the entire motoneuron or motoneuron terminal degeneration without death of the motoneuron.

Fig 1: X-axis label “Ag” “Age”?

Figure replaced as noted above.

78 See also: N. Arizono, O. Koreto, Y. Iwai, T. Hidaka, O. Takeoka, Morphometric analysis of human neuromuscular junctions in different ages, Acta Pathol Japon 34 (1984) 1243-1249.

Thank you for bringing this article to our attention. We have now included it, along with another reference to human NMJ morphology with aging. Please note that we are aware of one more work that provides morphological assessment of the human NMJ with aging (Jones et al. Cell Reports 21[9]: 2348-56, 2017); however, we have not included this for the following reason: the source human material used in this latter publication was almost exclusively from the amputated limbs of patients with long-term peripheral arterial disease (PAD) and it is known that recurring episodes of ischemia-reperfusion, as would occur in PAD patients, causes neuromuscular junction degeneration (e.g., Wilson et al. Free Rad Biol Med. 117: 180-90, 2018). Hence, we regard the data and conclusions from the Jones et al. work to be severely compromised by the source material. If the reviewer feels it worthwhile, we could include this work along with text explaining our concerns in the revised manuscript.

153 “Mitochondria in [aged?] skeletal muscle have…”

 Thank you for pointing out this error in the text.

Early in the manuscript it would be helpful for readers who do not work on mitochondria to define what is meant by “permeability transition” and give an idea of how this is usually detected/measured.

 This is now defined at the first mention of mitochondrial permeability transition in section 3.1

210-217 Fragmentation of the endplate AChR plaque is not a measure of NMJ impairment. It can result from muscle fibre regeneration as shown by the late Wesley Thompson and does not necessarily imply any impairment of NMJ function as Clarke Slater has shown (cited earlier in the manuscript). Lets not perpetuate this discredited marker. The Dupuis paper presents better evidence of NMJ impairment.

 As the reviewer correctly points out, we specify that most of the morphological alterations in the neuromuscular junction that occur with aging are likely of little functional consequence, based upon Clarke Slater’s data showing that neuromuscular junctions with fragmented AChRs have normal neuromuscular transmission characteristics. In the section of text referred to by the reviewer we make the point that over-expressing UCP1 in muscle causes alterations in neuromuscular junction morphology and specify AChR fragmentation as one of these changes. There is no implication of impaired function based upon the fragmentation. To avoid confusion, we now also refer to data from the Dupuis study showing that UCP-1 over-expression induced motoneuron terminal degeneration and hope this addresses the reviewer’s concern.

226 “…terminals [that?] can cause…”

 We have re-read this statement and believe the original wording conveys our meaning.

l.341-345 Please break this. difficult to read, sentence up. I can’t follow the meaning.

We have revised this statement as requested.